# Agave Syrup as an Alternative to Sucrose in Muffins: Impacts on Rheological, Microstructural, Physical, and Sensorial Properties

**DOI:** 10.3390/foods9070895

**Published:** 2020-07-08

**Authors:** César Ozuna, Eugenia Trueba-Vázquez, Gemma Moraga, Empar Llorca, Isabel Hernando

**Affiliations:** 1Departamento de Alimentos, División de Ciencias de la Vida, Campus Irapuato-Salamanca, Universidad de Guanajuato, Carretera Irapuato-Silao km 9, 36500 Irapuato, Guanajuato, Mexico; etruvaz@hotmail.com; 2Posgrado en Biociencias, División de Ciencias de la Vida, Campus Irapuato-Salamanca, Universidad de Guanajuato, Carretera Irapuato-Silao km 9, 36500 Irapuato, Guanajuato, Mexico; 3Grupo de Investigación Microestructura y Química de Alimentos, Departamento de Tecnología de Alimentos, Universitat Politècnica de València, Camino de Vera s/n, 46022 Valencia, Spain; gemmoba1@tal.upv.es (G.M.); emllomar@tal.upv.es (E.L.)

**Keywords:** inulin, bakery products, xanthan gum, leavening agent

## Abstract

Natural sweeteners, such as agave syrup, might be a healthy alternative to sucrose used in sweet bakery products linked to obesity. We evaluated the effect of sucrose replacement by agave syrup on rheological and microstructural properties of muffin batter and on physical and sensorial properties of the baked product. Muffins were formulated by replacing 25%, 50%, 75%, and 100% of sucrose by agave syrup (AS) and partially hydrolyzed agave syrup (PHAS), and by adding xanthan gum and doubled quantities of leavening agents. Rheological and microstructural properties of batter during baking were analyzed over the range of 25–100 °C. In the muffins, the structure, texture, color, and sensory acceptance were studied. The combination of agave syrup with xanthan gum and doubled quantities of leavening agents affected (*p* < 0.05) rheological and microstructural properties of the batters and textural properties of the low-sucrose muffins compared to the controls. The increase in agave syrup levels resulted in a darker crumb and crust. Sensory evaluation showed that AS-75 and PHAS-75 were the best alternatives to the control samples. Our results suggest a plausible substitution of up to 75% of sucrose by agave syrup in preparation of muffins, with physical and sensorial characteristics similar to those of their sucrose-containing counterparts.

## 1. Introduction

Nowadays, the eating habits of the world’s population include high ingestion of foods rich in sugars and fats, such as sweet bakery products [1,2]. Due to their practicality and pleasant taste, muffins are widely consumed by people of all ages, although mainly children [3,4]. The excessive consumption of these products has contributed to an increase in a series of non-communicable diseases and comorbidities, such as overweight, obesity [5], and dental caries [6]. According to the World Health Organization, almost 40% of the adult world population was overweight in 2016 and 13% were obese, whereas 40 million children under 5 years old were overweight or obese in 2018 [7].

In addition to providing sweetness to bakery products, sucrose contributes to the development of their structure, texture, and color [8]. Therefore, the replacement of sucrose content by artificial or natural sweeteners in the production of bakery products represents a challenge for the food industry [1,9,10]. Intense or non-caloric sweeteners (sucralose, saccharin, cyclamate, stevia, etc.) have great sweetening power; however, they do not contribute to the formation of the body of the bakery product [11]. On the other hand, energy sweeteners (monosaccharides, disaccharides, polyalcohols, etc.) can give rise to bakery products with stable structure, but they tend to lack in flavor [11]. Nevertheless, some natural agents may combine the best qualities of both groups of sweeteners: good sweetening power and a stable structure of the final bakery product; this group of sweeteners includes agave syrup (AS) [12,13,14].

Agave syrup is the sweet substance obtained by the hydrolysis of fructans present in the *Agave* spp. heads. In Mexico, where *Agave* spp. is endemic, there are about 205 species; however, agave syrup is obtained mostly from *Agave tequilana* Weber var. blue [15]. This natural sweetener, composed of fructose and fructooligosaccharides, has proven to have beneficial properties on human health, such as high prebiotic capacity and a low glycemic index score, and could prevent obesity and type II diabetes mellitus [16,17].

In bakery products, the use of agave syrup as a partial or total sucrose replacer has been employed in the elaboration of cookies [12], gluten-free cakes [13], and cereal bars [14]. However, the effects of agave syrup on the rheological, microstructural, and sensorial properties of bakery products remain unknown, both in the batter and in the final product.

The aim of this research was to evaluate the effects of sucrose replacement by agave syrup on rheological and microstructural properties of muffin batter, as well as on physical and sensorial properties of the baked product. Additionally, our study aimed to find added value in a natural sweetener that is currently underused in bakery products due to the lack of knowledge on its behavior during the baking process.

## 2. Materials and Methods

### 2.1. Muffin Batter Ingredients

The muffin batter ingredients included wheat flour (Cerealien Bischheim GmbH, Bischheim, Germany; composition provided by the supplier: 15% moisture, 12% protein); sugar (Pfeifer & Langen GmbH and Co., Cologne, Germany); whole liquid egg (Huevos Guillen S. L., Valencia, Spain); skimmed milk powder (Capsa Food, Asturias, Spain); refined sunflower oil (Sovena España S.A., Sevilla, Spain); agave syrup (Mieles Campos Azules S. A. de C. V., Amatitlan, Mexico; specifications of moisture and total sugars provided by the supplier: 23.20% moisture, 92.86% fructose, 0.15% glucose, 0.12% sucrose, 6.71% inulin, and 0.16% other carbohydrates); partially hydrolyzed agave syrup (PHAS) (Mieles Campos Azules S. A. de C. V., Amatitlan, Mexico; specifications of moisture and total sugars provided by the supplier: 23.30% moisture, 85.52% fructose, 0.40% glucose, 0.25% sucrose, 13.58% inulin, and 0.25% other carbohydrates); xanthan gum (Cargill France SAS, Puteaux, France); leavening agents, including sodium bicarbonate, malic acid, and tartaric acid (Hacendado, Valencia, Spain); and salt (Salinas Del Odiel, S. L., Huelva, Spain). The sucrose equivalent (SE) of agave syrup (AS) and partially hydrolyzed agave syrup (PHAS) was calculated using the equation proposed by Koehler and Kays [18] and Belščak-Cvitanović et al. [19]; the SE values for AS and PHAS were 1.61 and 1.49, respectively.

### 2.2. Batter and Muffin Preparation

Nine muffin batters were prepared according to batter formulations in Table 1. In the case of batters elaborated with sucrose replacement by AS and PHAS (25% replacement: AS-25 and PHAS-25; 50% replacement: AS-50 and PHAS-50; 75% replacement: AS-75 and PHAS-75; and 100% replacement: AS-100 and PHAS-100), the concentration of leavening agents was doubled and xanthan gum was used as a loading agent according to Martínez-Cervera et al. [3].

The all-in mixing procedure reported by Rodríguez-García et al. [20] was employed in batter preparation, with some modifications. First, the liquid ingredients (including 0.5 g of xanthan gum dissolved in 30 g of water), except for the sunflower oil, were introduced into the commercial kneader (Thermomix, TM31, Wuppertal, Germany) and mixed for 1 min at a speed of 200 rpm. Then, the solid ingredients were added into the same container and mixed for an additional 2 min at the same speed. Finally, sunflower oil was added and mixed in for 3 min at 500 rpm. Once the smooth batter was obtained, 45 g were placed in paper molds and subsequently deposited in silicone trays. Finally, they were introduced into an electric oven (Electrolux, EOC3430DOX, Stockholm, Sweden) that had been preheated to 180 °C. Samples were baked for 30 min at 180 °C. Each baking batch consisted of 12 muffins and each formulation was carried out in triplicate, representing 36 samples per formulation. All analyses were performed 24 h after baking in order to ensure stability in the samples.

### 2.3. Rheology and Microstructure of Batters

Rheological and microstructural analyses were performed in batters with 0% (control), 50%, and 100% sucrose replacement by both AS and PHAS. Rheological measurements were carried out using a rotational rheometer (Kinexus Pro+, Malvern Panalytical, Malvern, UK) equipped with a Peltier plate cartridge. A series of tests were performed at 20 °C with parallel plate geometry (Φ = 40 mm), with the geometry gap set at 1500 µm. Before the rheological test, the batters were all kept at 25 °C for 60 min post-preparation. Flow measurements were conducted (shear rate 1 s^−1^ to 100 s^−1^, temperature = 25 °C), along with frequency sweeps (stress = 10 Pa, frequency = 0.1–10 Hz, temperature = 25 °C) and temperature sweeps in the linear viscoelastic region (frequency = 1 Hz, stress = 100 Pa, temperature = 25–100 °C). Vaseline oil was applied to the exposed surfaces of the samples to prevent sample drying during testing.

Microscopical examination of muffin batters during simulated micro-baking was carried out following the methodology proposed by Rodríguez-García et al. [20]. One drop of the sample was placed in the concavity of a glass slide and placed into a temperature-controlled vault. The temperature in the vault rose steadily from 25 °C to 105 °C at the rate of 1.5 °C/min. Batter samples were observed at 4× magnification (objective lens 4 × /0.13∞/ − WD 17.1, Nikon, Tokyo, Japan). A camera (ExWaveHAD, model no. DXC-190, Sony, Tokyo, Japan) was attached to the microscope and connected to the video entry port of a computer. Images were captured and stored in the format of 640 × 540 pixels using the microscope software (Linksys 32, Linkam, Surrey, UK).

### 2.4. Muffin Height and Crumb Structure

Muffins were cut vertically with a stainless steel knife and scanned by means of a conventional scanner (Epson Perfection 1250; Epson America, Inc., Long Beach, CA, USA) at a resolution of 300 dpi. The maximum muffin height and the crumb structure were measured using ImageJ software (National Institute of Health, Bethesda, MA, USA). Each image was cropped to a 30 × 30 mm section on which the crumb analysis was performed. The image was split into color channels, the contrast was enhanced, and the image was binarized at the grey scale, resulting in air cells being colored in black and the rest of the crumb in white. Four macroscopic characteristics of the crumb cell structure were calculated, namely the cell area (mm^2^), cell circularity, cell density, and total cell area within the crumb (%). The data for the muffin height and crumb structure were obtained from 18 different images for each formulation.

### 2.5. Muffin Texture

Muffin textural properties were evaluated using a texture analyzer (Stable Micro System, TA-XT plus, Godalming, UK) and the Texture Exponent Lite 32 program (version 6.1.4.0, Stable Micro Systems, Godalming, UK). For texture profile analysis (TPA), cubes were cut from the central area of the muffin (1.5 cm per side). Double compressions of 40% deformation at a speed of 1 mm/s were performed, with a resting time of 5 s between the two compressions. Compression was performed with a 35-mm diameter aluminum plate. After the two compression cycles, the following parameters were recorded: hardness, elasticity, cohesiveness, and chewiness [21].

### 2.6. Muffin Crust and Crumb Color

Color measurements of both the muffin crust and crumb were carried out with a CR-400 chroma meter (Konica Minolta Sensing Americas, Inc., Ramsey, MN, USA). The results were expressed in accordance with the International Commission on Illumination (CIELAB) system with reference to illuminant C and a visual angle of 2°. The determined parameters were *L** (lightness, *L** = 0 (black) and *L** = 100 (white)), *a** ((*a** negative values = greenness and *a** positive values = redness)), and *b** ((*b** negative values = blueness and *b** positive values = yellowness)). The chroma (*C**_*ab*_), hue angle (*h**_*ab*_), and total color difference (Δ*E**) were calculated using Equations (1)–(3):(1)$$C_{ab}^{*}=\sqrt{a^{*2}+b^{*2}}$$
(2)$$h_{ab}^{*}=tan^{-1}\frac{\left(b^{*}\right)}{\left(a^{*}\right)}$$
(3)$$∆E^{*}=\sqrt{{\left(L^{*}-L_{0}^{*}\right)}^{2}+{\left(a^{*}-a_{0}^{*}\right)}^{2}+{\left(b^{*}-b_{0}^{*}\right)}^{2}}$$
where $L_{0}^{*}$, $a_{0}^{*}$, and $b_{0}^{*}$ represent the values of the chromatic coordinates of the control muffin.

### 2.7. Sensory Evaluation of Final Product

Seventy untrained judges (18–80 years old, recruited from the Life Sciences Division of the University of Guanajuato, Irapuato, Mexico) simultaneously evaluated the sensory characteristics (appearance, flavor, texture, color, and general acceptability) of the control and the eight muffin formulations made with AS and PHAS. The product acceptability was determined using a nine-point hedonic scale (9 = like; 1 = dislike) [22].

### 2.8. Statistical Analysis

Two-way analyses of variance (ANOVAs) with sucrose replacement (0%, 25%, 50%, 75%, and 100%) and agave syrup type (AS and PHAS) as between-subject factors were carried out on all dependent variables in SPSS 18.0 software (SPSS Inc., Chicago, IL, USA). Tukey post-hoc tests were performed for all significant main effects and interactions.

## 3. Results and Discussion

### 3.1. Rheological Properties of Muffin Batters

Figure 1 shows the effect of sucrose replacement (at 0%, 50%, and 100% reduction levels) by AS and PHAS and a combination of xanthan gum and doubled quantities of leavening agents on flow curves of the muffin batters.

A typical shear thinning behavior was observed in all batters (Figure 1). Despite having used a combination of xanthan gum and doubled quantities of leavening agents, the batters were less viscous when AS and PHAS were employed as sucrose replacers. This behavior is quite possibly related to the moisture content of agave syrups (23.20% and 23.30% for AS and PHAS, respectively) which would have induced the decrease in AS and PHAS batter viscosities. At a shear rate of 60 s^−1^, the apparent viscosity significantly (*p* < 0.05) decreased as more sucrose was replaced by AS and PHAS, from 3.35 (0.01; the values in parentheses refer to the standard deviation of the mean) Pa·s in the control sample to 2.48 (0.04) Pa·s and 1.98 (0.04) Pa·s in 50% and 100% sucrose-replaced batters, respectively. There was no significant (*p* > 0.05) effect of agave syrup type, nor an interaction between the two factors. Other authors also stated the decrease in the batter viscosity due to the sucrose replacement by syrups resulting in a low product volume and poor cell structure [2].

Figure 2A,B show the frequency dependence of the different batters on the elastic modulus (G′), the viscous modulus (G″), and the phase angle (δ) at 25 °C. At low frequencies (0.1–1 Hz), the control batter behaved as a solid (G′ > G″; δ < 45), while at high frequencies (>1 Hz) this batter exhibited a liquid-like behavior (G′ < G″; δ > 45). On the other hand, batters formulated with AS (AS-50 and AS-100) and PHAS (PHAS-50 and PHAS-100), using a combination of xanthan gum and doubled quantities of leavening agents, showed a solid-like behavior (G′ > G″; δ < 45), regardless of the frequencies tested, caused by the gelling effect of xanthan gum in the batter. At 1 Hz of frequency, sucrose replacement significantly (*p* < 0.05) affected the G′ modulus of batters, with higher sucrose substitution levels producing higher G′ values. However, there was no significant effect of agave syrup type (*p* > 0.05), nor an interaction between the two factors.

The competition for the available water between sucrose and xanthan gum led to a less elastic semi-solid network, with the G′ values obtained for batters with 0, 50, and 100% of sucrose replacement being 42.54 (5.43), 84.24 (4.90), and 98.42 (10.55) Pa, respectively. On the other hand, the G″ modulus was not significantly affected by the factors studied or their interaction (all *p*s > 0.05). At 1 Hz, the G″ values ranged from 45.26 (2.09) to 46.63 (1.95) Pa for the control and the 100% sucrose replacement batter, respectively. As for the δ values, these were significantly affected by sucrose substitution (*p* < 0.05), with higher sucrose replacement levels producing lower δ values, indicating a more solid-like behavior. The actual δ values obtained at 1 Hz for 0, 50, and 100% sucrose replacement were 46.98° (2.35), 28.31° (0.45), and 25.50° (1.68), respectively. There was no significant main effect of agave syrup type (*p* > 0.05), nor an interaction between the two factors.

To analyze the structural changes provoked by heat in muffin batters, linear viscoelastic properties were studied in the range of 25–100 °C in order to simulate batter behavior during baking. The effects of temperature on the complex modulus G* and δ during batter heating are shown in Figure 2C,D, respectively. The G* modulus, a measure of batter stiffness, was lower in the control sample compared to the samples with sucrose replacement along all the sweep temperatures tested. As the temperature increased, batter stiffness increased as well; this was probably caused primarily by starch gelatinization and protein denaturation [23]. The temperature at which the G* value started to increase was 60 °C for the control batter, however it shifted to 75 °C and 85 °C as sucrose was replaced at 100% and 50%, respectively. In this sense, a synergic effect between sucrose and xanthan gum was observed, delaying the starch gelatinization, and the thermosetting temperature increased in batters with 50% replacement. Most probably, the moisture content of the agave syrups contributed to higher values of the thermosetting temperature in replaced muffins than in the control, since a higher water content needed to be evaporated for the structure of the replaced muffins to be built. A decrease in δ values above those temperatures was also detected, reflecting the predominance of the elastic behavior versus the viscous behavior during starch gelatinization and protein denaturation [24]. In summary, the sucrose replacement by AS and PHAS and a combination of xanthan gum and doubled quantities of leavening agents increased the thermosetting temperature in batters. This fact is important for the correct formation of water vapor and CO_2_ in the batter, as well as their diffusion and expansion into occluded air cells during the baking process [3,25].

### 3.2. Microstructural Properties of Muffin Batters

Figure 3A shows the effect of sucrose replacement (at 0%, 50%, and 100% reduction levels) by AS and PHAS in combination with the addition of xanthan gum and doubled quantities of leavening agents on batter microstructures during simulated micro-baking. These microstructural images were analyzed to quantify the bubble area (Figure 3B).

When the temperature increased, there was an expansion of the bubbles in all samples and the bubble size distribution tended to widen (Figure 3A). In the control batter, the number of CO_2_ bubbles was higher if compared to replaced batters and the bubble size increased at a controlled and uniform rate. This behavior may have been due to the high viscosity of this batter (as observed previously in Figure 1), which may have reduced bubble movement in the batter and slowed down the coalescence phenomena. Thus, regardless of temperature, the control batters showed a higher frequency of small bubble sizes (0–10,000 μm^2^) in comparison with replaced batters (Figure 3B).

On the other hand, a lower viscosity of the replaced batters could have aided in bubble coalescence, giving place to bigger bubbles. When sucrose was totally replaced, a high frequency of big bubble sizes (over 60,000 μm^2^) could be observed for AS-100 and PHAS-100 when compared to the other batters. Regarding the influence of agave syrup type, PHAS samples showed a higher frequency of big bubbles (over 140,000 μm^2^) than AS samples at both 50% and 100% sucrose replacement (Figure 3B).

### 3.3. Muffin Crumb Structure

Figure 4 shows the effect of sucrose replacement by AS and PHAS on the macroscopic characteristics of the crumb cell structure, measured by image analysis.

Regarding cell density, the main effect of the sucrose replacement was significant (*p* < 0.05), with lower cell density for samples with 25% and 50% sucrose substitution in comparison to control and samples with 75% and 100% sucrose substitution (Figure 4A). There was a significant main effect (*p* < 0.05) of agave syrup type, which did not interact with sucrose replacement (*p* > 0.05). The main effect of sucrose replacement on cell area was significant (*p* < 0.05). As sucrose substitution increased, the average air cell size was higher than the control muffin (Figure 4B). There was also a significant main effect (*p* < 0.05) of agave syrup type, which did not interact with sucrose replacement (*p* > 0.05). A higher average cell area was obtained when substituting sucrose by PHAS than by AS.

As for total cell area (%), there were significant main effects (*p* < 0.05) of sucrose replacement and agave syrup type, with higher levels of sucrose replacement and PHAS reaching higher percentages of total cell area than control muffin (Figure 4C). There was a significant interaction (*p* < 0.05) between the effects of the two variables, caused by an inversion in the general trend at 75% of sucrose substitution by PHAS. Sucrose replacement affected cell circularity, but a significant (*p* < 0.05) reduction was only observed for the formulations with 100% substitution (Figure 4D). However, there was no main effect of agave syrup type on cell circularity, nor an interaction between the two factors (*p* > 0.05).

The obtained results are in accordance with observations of the bubble expansion during micro-baking (Figure 3), where a high frequency of small bubble sizes in control batters could be observed. The variations that occurred in the muffin crumb as sucrose substitution increased are due mainly to the fact that with the addition of xanthan gum and doubled quantity of leavening agents, the batters with sucrose substitution comprised a greater amount of gas. In addition, the low viscosity of the batters allowed for the gas bubbles to have more mobility, provoking their coalescence. The gases did not reach the surface since their exit was hindered by an early formation of the crust, resulting in the setting of the bubbles in the form of diffusion pathways. Thus, the samples with over 25% sucrose replacement by both AS and PHAS reached a height of over 57.33 (2.10) mm. When compared to the average height of 50.93 (2.27) mm obtained from the control samples, significant (*p* < 0.05) differences in height can be observed for all samples with sucrose replacement by AS and PHAS.

### 3.4. Muffin Texture

Figure 5 shows the effect of sucrose replacement by AS and PHAS on the texture profile analysis of the final products. Since the textural properties of muffins depend greatly on their crumb structure, all samples with over 25% sucrose substitution by either AS or PHAS presented significant differences (*p* < 0.05) with the control for all textural properties evaluated. As a result, all the replaced muffins considerably differed from the control regardless of syrup type or replacement level.

Regarding hardness, the main effect of sucrose replacement was significant (*p* < 0.05), with lower hardness values recorded for samples with sucrose replacement than control, decreasing from 1.27 (0.30) N in the control sample to a range of 0.65 (0.27) to 0.85 (0.32) N for the rest of the samples. Even though agave syrup type did not have a significant main effect (*p* > 0.05) on this attribute, it did interact with sucrose replacement (*p* < 0.05) (Figure 5A).

The main effect of sucrose replacement on springiness was significant (*p* < 0.05). As sucrose substitution increased, the average values tended to increase as well. There was also a significant main effect of agave syrup type on springiness (*p* < 0.05), which did not interact with sucrose replacement (*p* > 0.05). Moreover, a significant difference (*p* < 0.05) was observed between the control sample and the formulations with over 25% sucrose substitution, where the values increased from 0.32 (0.03) in the control formulation to a range of values between 0.38 (0.02) and 0.41 (0.02) for the other formulations. On the other hand, slightly higher springiness values were obtained when substituting sucrose with AS than PHAS (Figure 5B).

Cohesiveness was significantly affected (*p* < 0.05) by sucrose replacement, as well as by agave syrup type. However, there was no significant interaction between these variables (*p* > 0.05). As sucrose substitution levels escalated, cohesiveness significantly increased (*p* > 0.05) as well, from 0.71 (0.02) in the control formulation to at least a value of 0.74 (0.02) for the other samples. Similarly to springiness, slightly higher cohesiveness values were obtained when substituting sucrose with AS than PHAS (Figure 5C). As for chewiness, the results are very similar to hardness. The main effect of sucrose replacement was significant (*p* < 0.05), while the main effect of agave syrup type was not (*p* > 0.05). There was a significant interaction between the effect of sucrose replacement and agave syrup type (*p* < 0.05) caused by an inversion in the general trend at 50% sucrose substitution by AS (Figure 5D).

In replaced batters, the gases were not able to fully exit the product due to the increase in the amount of gas production during baking, which was caused by the doubling of leavening agents and the low permeability of the crust. Along with this, the low viscosity of the batter allowed for the air bubbles to move and coalesce, causing the muffin crumb to contain large air cells resembling diffusion pathways, as well as compact areas [26]. This particular structure of the crumb led to increases in cohesiveness and springiness, due to the fact that the presence of air cells allowed the muffins to easily recover their original shape and size after being compressed.

Regarding the decrease in hardness shown by the sucrose-substituted muffins, this was probably caused by the increase in total cell area (%). The products became softer because the air cells did not give any resistance during their first compression. Since chewiness refers to the difficulty to chew and create food bolus, its value depends on the product hardness, which is why the products were easier to chew as they became less hard [3]. The decrease in muffin chewiness could be also attributed to the increase in cohesiveness, since this characteristic allows for the formation of a bolus instead of fracturing or crumbling upon mastication [27]. According to Gao et al. [28], partial and total replacement of sucrose in sweetened baked products usually results in higher hardness and chewiness values, as well as in lower springiness. This is credited to the ability of sucrose to delay starch gelatinization. Nonetheless, due to the incorporation of double quantities of leavening agents, our muffins showed opposite results [29].

### 3.5. Muffin Crust and Crumb Color

In general terms, the control muffin showed a golden-brown crust and a yellow crumb, which are both characteristic of bakery products. However, as the percentage of sucrose substitution increased, the muffin color in general tended to be darker than in control samples, with the muffin crust turning towards red hues and the crumb losing some of its yellowness.

The main effect of sucrose replacement on *L** was significant (*p* < 0.05) in both the muffin crust and crumb. As sucrose substitution increased, the *L** values (ranging from 46.27 to 42.50 for the crust and from 61.34 to 53.23 for the crumb) significantly decreased (*p* < 0.05) in comparison with the control samples (58.81 and 69.16 for the crust and crumb, respectively).

Regarding the *C**_*ab*_ of the crust, significant effects (*p* < 0.05) were observed for sucrose replacement, as well as for agave syrup type, with a significant interaction (*p* < 0.05) between these factors. As for the crumb *C**_*ab*_, only sucrose substitution had a significant effect (*p* < 0.05). As sucrose replacement increased, the color intensity of both the crust and crumb of the muffins showed different tendencies; while the crusts values decreased (40.59 for control and 35.99–32.22 for replaced muffins), the crumb values increased (21.87 for control and 30.27–27.50 for replaced muffins). Both presented significant differences (*p <* 0.05) between the control and the rest of the samples.

The main effect of sucrose substitution on *h**_*ab*_ was significant (*p* < 0.05) on both the muffin crust and crumb. Replaced muffins exhibited lower *h**_*ab*_ values of both the crumb and crust (ranging from 65.02 to 62.47 for the crust and from 84.07 to 76.10 for the crumb) than control samples (77.55 and 94.79 for the crust and crumb, respectively). Moreover, as percentages of sucrose replacement increased, the crumb *h**_*ab*_ values decreased. This shows that sucrose replacement by agave syrup provokes hue changes from yellow to orange-red, both in the muffin crumb and crust.

As for Δ*E**, the main effect of the two factors was significant (*p* < 0.05) for the crust and the crumb. In general, the samples in which sucrose was replaced with PHAS syrup showed higher Δ*E** values in the crust (20.45), whereas crumb Δ*E** values were similar for both AS and PHAS (ranging from 14.66 to 18.50). According to Bodart et al. [30], when the values obtained for total color difference are over three, the color variations between the analyzed samples are visible to the human eye. In this study, all samples with sucrose replacement showed Δ*E** values of over three when compared to the control formulation, meaning the variations in color were easily noticeable to the naked eye.

The differences in color exhibited by the different formulations can mainly be attributed to the intensification of non-enzymatic browning (Maillard reaction) that happened as the amount of incorporated agave syrup increased [31,32]. Similar results were reported by Kocer et al. [33]—as the amount of sucrose replaced by polydextrose increased, the baked products became darker. When substituting sucrose by other sweeteners such as erythritol, sorbitol, maltitol, xylitol, and mannitol, the opposite tends to occur, as these sweeteners do not contribute to the Maillard reaction [3,34]. On a smaller scale, the muffin color was also affected by the dark color of the syrups in comparison to sucrose. Temperature fulfills an important role regarding the color of the crust and crumb of bakery products. Since the Maillard reaction is temperature-dependent, its effect will vary according to the maximum temperature reached. The crust tends to present a greater change due to the fact that it is exposed to higher temperatures, while the crumb’s exposure to the heat is more limited [32].

### 3.6. Sensory Evaluation of Final Product

Figure 6 shows the effects of sucrose replacement by AS and PHAS on sensory acceptance of muffins. The main effect of sucrose replacement on the appearance and flavor of the final products (Figure 6A,B, respectively) was significant (*p* < 0.05). There was also a significant main effect of agave syrup type (*p* < 0.05), which interacted with sucrose replacement (*p* < 0.05). Significant differences (*p* < 0.05) between AS and PHAS were observed at 100% sucrose substitution. Throughout all samples, PHAS maintained higher average scores than AS. The interaction in the case of appearance was caused by the decrease of scores that occurred at 50% sucrose substitution by PHAS, the point at which the tendency for the samples with AS kept rising.

Muffin texture was significantly affected by sucrose substitution (*p* < 0.05), as well as by agave syrup type (*p* < 0.05). However, there was no significant interaction between these factors (*p* > 0.05). The texture of samples with AS and PHAS had a similar tendency for all substitution levels, with muffins made with 25% and 50% sucrose substitution yielding significantly (*p* < 0.05) better texture acceptance than the control and 75% and 100% substitution levels (Figure 6C). In general, PHAS had better texture acceptance than AS.

Regarding color and general acceptance, the main effects of both sucrose replacement and agave syrup type were significant (*p* < 0.05). There was a significant interaction between these factors (*p* < 0.05). Although the samples where sucrose was substituted by PHAS did not show significant differences (*p* < 0.05) between replacement levels, the muffins with AS showed less consistency in their scores, presenting significant differences (*p* < 0.05) for the samples with 100% sucrose substitution.

As sucrose substitution increased, the muffin color became darker. This factor favored the color score up to 75% of sucrose substitution by AS. Some judges singled out the fact that the darker color gave the muffins a more “rustic” and “home-made” appearance, which they liked. However, once the color became too dark, the change became unfavorable. On the other hand, when PHAS was used as sucrose replacement, there were no significant differences in the color score of the different samples (Figure 6D). Taking all aspects into consideration, it is possible to say that the samples made by substituting 75% of the sucrose with either AS or PHAS syrups were the best alternative to the control samples (Figure 6E).

## 4. Conclusions

Based on the results of our study, both AS and PHAS can be used as alternatives for sucrose in muffin formulations in combination with the addition of xanthan gum and doubled quantities of leavening agents. The sucrose replacement by AS and PHAS reduced the batter viscosity and increased the thermosetting temperature in batters from 60 °C for the control to 75–85 °C for replaced batters. The textural properties of muffins depend greatly on their crumb structure; all the replaced muffins considerably differed from the control regardless of syrup type or replacement level. As for color properties, as the percentage of sucrose substitution increased, muffin color in general tended to be darker than in control samples, with the muffin crust turning towards red hues and the crumb losing some of its yellowness. Even though the percentage of sucrose replacement affected the rheological and microstructural properties of the batters and physical parameters analyzed in the baked products, the sensory evaluation of the muffins suggested that both types of agave syrup can be good alternatives up to 75% sucrose substitution. When 100% of the sucrose was replaced, muffins formulated with PHAS showed better texture, flavor, color, and general acceptability than those formulated with AS.

## Figures and Tables

**Figure 1 foods-09-00895-f001:**
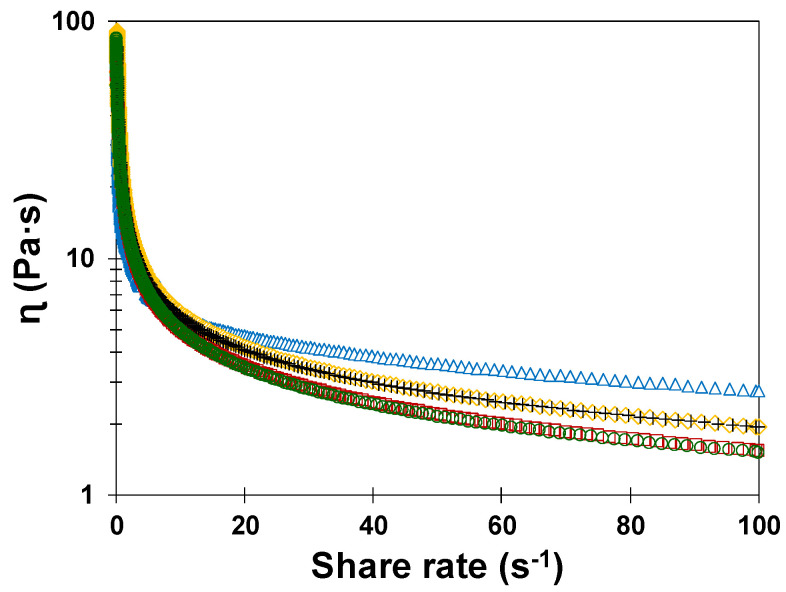
The impact of sucrose replacement by agave syrup (AS) and partially hydrolyzed agave syrup (PHAS) on flow curves of the muffin batters: control (blue triangles), PHAS-50 (black crosses), AS-50 (yellow diamonds), PHAS-100 (green circles), and AS-100 (red squares).

**Figure 2 foods-09-00895-f002:**
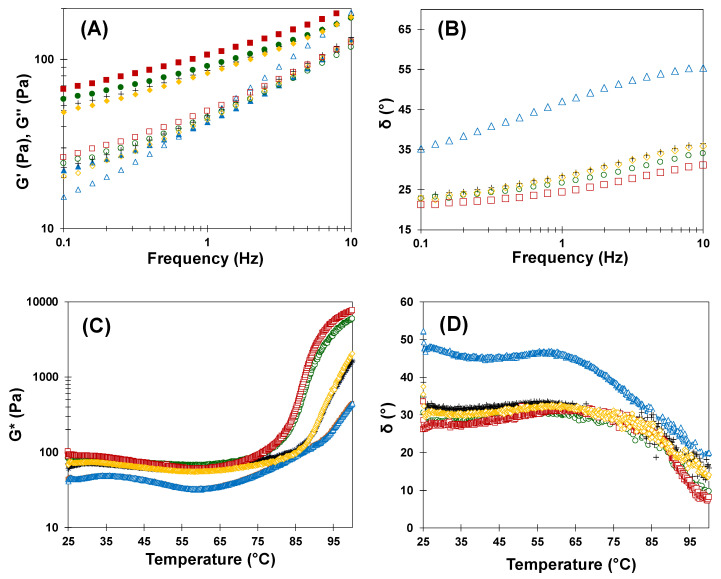
The impact of sucrose replacement by AS and PHAS on frequency (**A**,**B**) and temperature (**C**,**D**) sweeps of the muffin batters: control (blue triangles), PHAS-50 (black crosses), AS-50 (yellow diamonds), PHAS-100 (green circles), and AS-100 (red squares). In Figure 2A, two variables are shown: G′ (full data points) and G″ (empty data points).

**Figure 3 foods-09-00895-f003:**
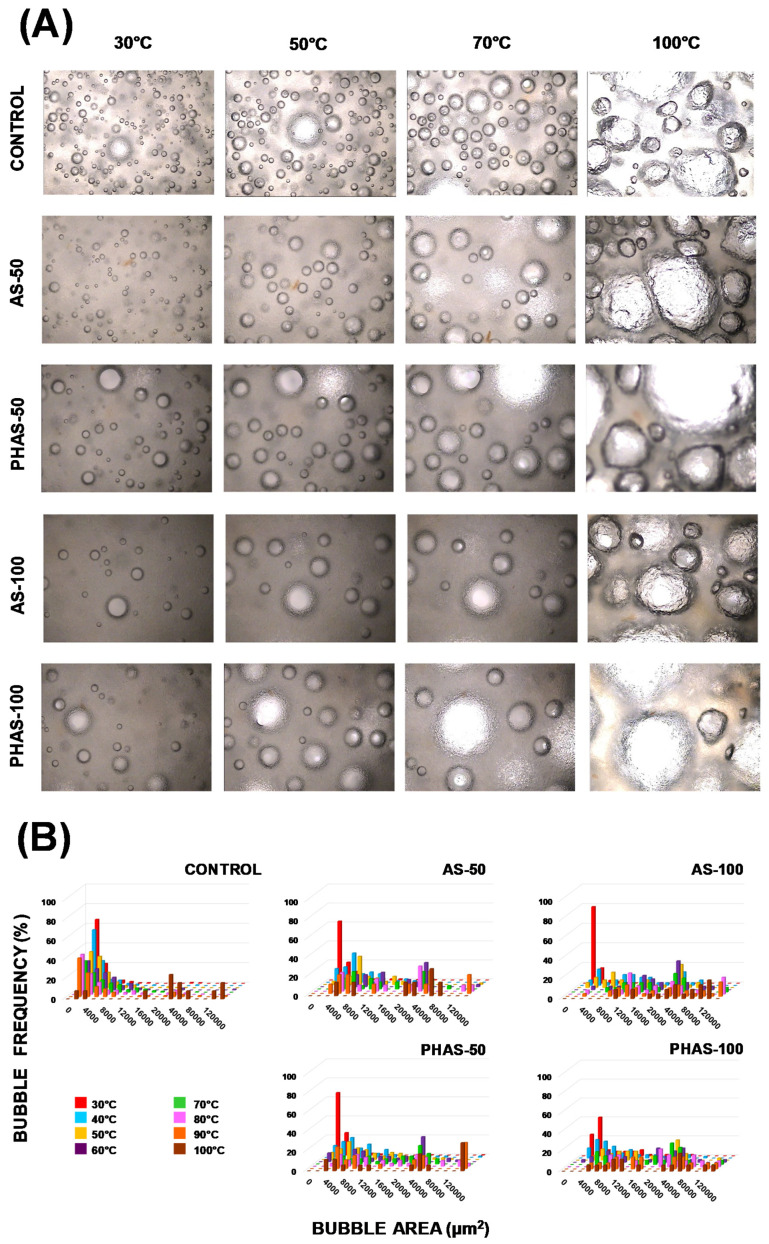
Changes in batters during micro-baking by baking temperature: (**A**) light microscopy images (4×) of bubble expansion; (**B**) bubble size distribution histograms.

**Figure 4 foods-09-00895-f004:**
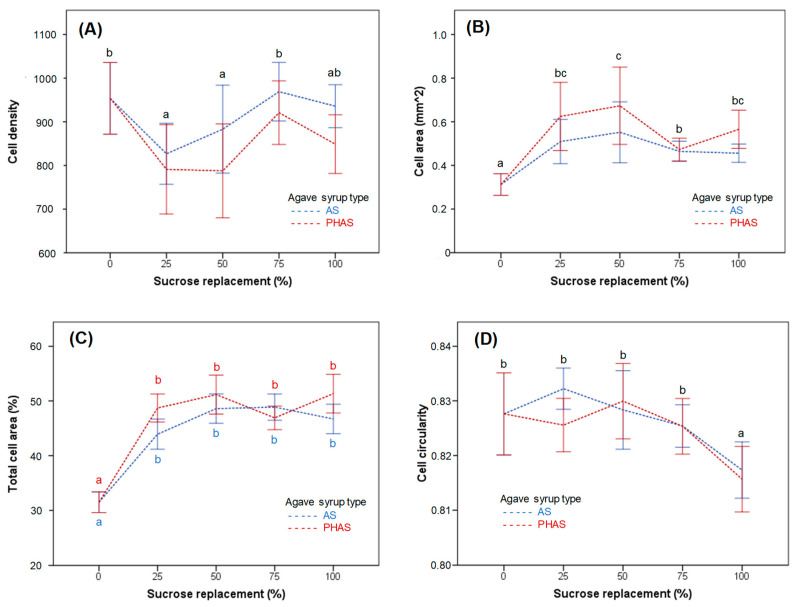
The impact of sucrose replacement by AS and PHAS on the muffin crumb structure, namely cell density (**A**), cell area (**B**), total cell area (**C**), and circularity (**D**). Error bars represent 95% confidence intervals. The different subscript letters of the same color represent Tukey’s homogeneous groups, with significant (*p* < 0.05) differences between different sucrose replacement levels for AS (blue letters), PHAS (red letters), or for both types of agave syrup when there was no interaction between the sucrose replacement and agave syrup type (black letters).

**Figure 5 foods-09-00895-f005:**
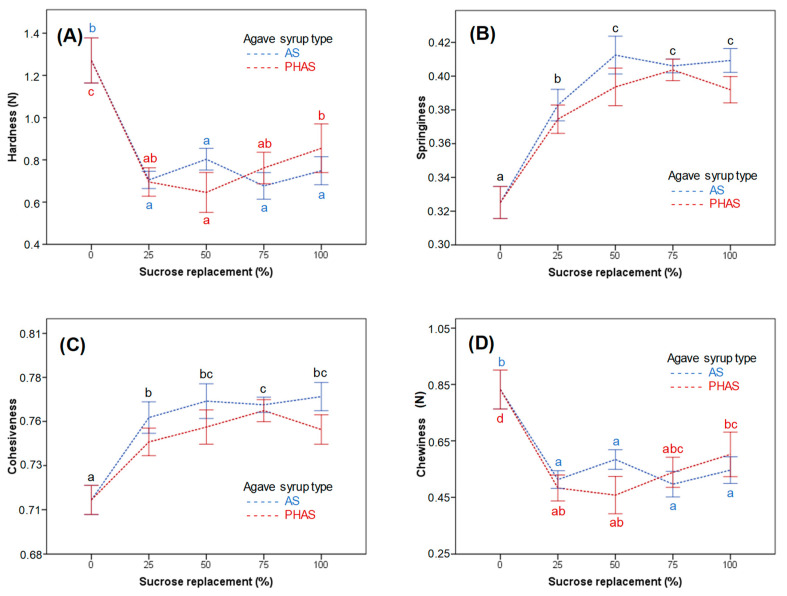
The impact of sucrose replacement by AS and PHAS on the textural profile analysis of muffins: (**A**) hardness, (**B**) springiness, (**C**) cohesiveness, and (**D**) chewiness. The different subscript letters of the same color represent Tukey’s homogeneous groups, with significant (*p* < 0.05) differences between different sucrose replacement levels for AS (blue letters), PHAS (red letters), or for both types of agave syrup when there was no interaction between sucrose replacement and agave syrup type (black letters).

**Figure 6 foods-09-00895-f006:**
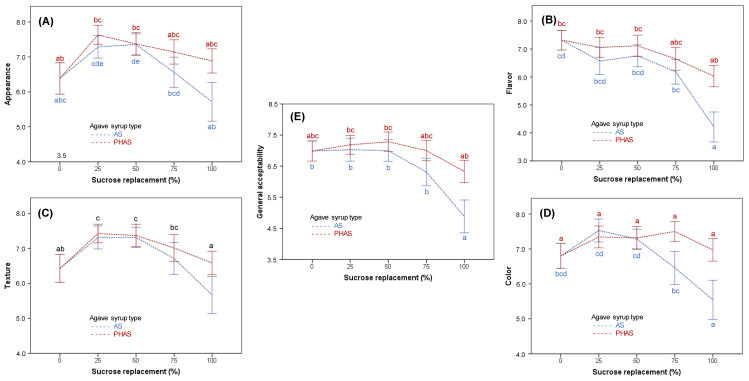
The impact of sucrose replacement by AS and PHAS on sensory properties in muffins, namely appearance (**A**), flavor (**B**), texture (**C**), color (**D**), and general acceptability (**E**). The letters above each data point represent Tukey’s homogenous groups (*p* < 0.05) calculated for the sucrose replacement factor. Error bars represent 95% confidence interval. The different subscript letters of the same color represent Tukey’s homogeneous groups, with significant (*p* < 0.05) differences between different sucrose replacement levels for AS (blue letters), PHAS (red letters), or for both types of agave syrup when there was no interaction between sucrose replacement and agave syrup type (black letters).

**Table 1 foods-09-00895-t001:** Formulations of the different muffin batters (percentage on wheat flour basis).

Ingredients	CONTROL	AS-25	AS-50	AS-75	AS-100	PHAS-25	PHAS-50	PHAS-75	PHAS-100
Wheat flour	100	100	100	100	100	100	100	100	100
Sucrose	100	75	50	25	-	75	50	25	-
AS	-	15.48	30.95	46.42	61.90	-	-	-	-
PHAS	-	-	-	-	-	16.68	33.36	50.05	66.73
Xanthan gum	-	0.5	0.5	0.5	0.5	0.5	0.5	0.5	0.5
Pure water	-	30	30	30	30	30	30	30	30
Whole liquid egg	81	81	81	81	81	81	81	81	81
Skimmed milk	100	100	100	100	100	100	100	100	100
Sunflower oil	46	46	46	46	46	46	46	46	46
Sodium bicarbonate	4	8	8	8	8	8	8	8	8
Malic and tartaric acid	3	6	6	6	6	6	6	6	6
Salt	1.5	1.5	1.5	1.5	1.5	1.5	1.5	1.5	1.5

AS: Agave syrup; PHAS: Partially hydrolyzed agave syrup.

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
