# Peer review of "Agave Syrup as an Alternative to Sucrose in Muffins: Impacts on Rheological, Microstructural, Physical, and Sensorial Properties"

_foods, 2020, doi:10.3390/foods9070895_

Round 1

Reviewer 1 Report

This study tests effect of sucrose replacement by agave syrup on rheological, microstructural and sensorial properties of muffins. The introduction provides a general background on the role of sugars (especially sucrose) in bakery products The objective of the study is clearly defined.

The experimental apparatus is standard and is appropriate for the study. The methods are well described and provide sufficient information to reproduce the experiments. The results are clearly explained and presented in an appropriate format.

The conclusions should be more specific. Please state the effect of the syrup on the rheological properties of the batter as well as on the textual and sensory properties of muffins.

Additional comments for Authors:

  1. In line 172, explain what the values in parentheses mean.

“At a shear rate of 60 s-1, the apparent viscosity significantly (p<0.05) decreased as more sucrose was replaced by AS and PHAS, ranging from 3.35 (0.01) Pa.s in control sample to 2.48 (0.04) Pa.s and 1.98 (0.04) Pa.s in 50% and 100% sucrose-replaced batters, respectively.

  1. Change the viscosity unit notation from Pa.s to Pa·s

Author Response

Reviewer comments:

REVIEWER 1

This study tests effect of sucrose replacement by agave syrup on rheological, microstructural and sensorial properties of muffins. The introduction provides a general background on the role of sugars (especially sucrose) in bakery products. The objective of the study is clearly defined.

The experimental apparatus is standard and is appropriate for the study. The methods are well described and provide sufficient information to reproduce the experiments. The results are clearly explained and presented in an appropriate format.

  1. COMMENT: The conclusions should be more specific. Please state the effect of the syrup on the rheological properties of the batter as well as on the textual and sensory properties of muffins.

ANSWER: We agree with the reviewer. Your suggestions have been incorporated into the conclusion section.

Revised manuscript (Lines 417-430):

Based on the results of our study, both AS and PHAS could be used as an alternative for sucrose in muffin formulations, in combination with the addition of xanthan gum and doubled quantities of leavening agents. The sucrose replacement by AS and PHAS reduced the batter viscosity and increased the thermosetting temperature in batters from 60°C for the control to 75‑85°C for replaced batters. The textural properties of muffins depend greatly on their crumb structure and all the replaced muffins considerably differed from the control regardless of syrup type or replacement level. As for color properties, as the percentage of sucrose substitution increased, muffin color in general tended to be darker than in control samples, with the muffin crust turning towards red hues and the crumb losing some of its yellowness. Even though the percentage of sucrose replacement affected the rheological and microstructural properties of the batters and physical parameters analyzed in the baked products, the sensory evaluation of the muffins suggested that both types of agave syrup could be a good alternative up to 75% of sucrose substitution. When 100% of sucrose was replaced, muffins formulated with PHAS showed better texture, flavor, color, and general acceptability than those formulated with AS.”

Additional comments for Authors:

  1. COMMENT: In line 172, explain what the values in parentheses mean.

“At a shear rate of 60 s-1, the apparent viscosity significantly (p<0.05) decreased as more sucrose was replaced by AS and PHAS, ranging from 3.35 (0.01) Pa.s in control sample to 2.48 (0.04) Pa.s and 1.98 (0.04) Pa.s in 50% and 100% sucrose-replaced batters, respectively.

ANSWER: In the entire manuscript, the values in parentheses refer to the standard deviation of the mean. This was added as a footnote on page 5.

  1. COMMENT: Change the viscosity unit notation from Pa.s to Pa·s

ANSWER: The viscosity unit notation has been changed in Figure 1 and in the revised manuscript.

Revised manuscript (Line 175):

from 3.35 (0.011) Pa·s in control sample to 2.48 (0.04) Pa·s and 1.98 (0.04) Pa·s in 50% and 100%”

Reviewer 2 Report

This study was more like a product development work, how to replace sucrose levels with Agave syrup and with help of a hydrocolloid. The pre tests had been done as the levels of xanthan and agents were known. The characteristics of the muffins of Agave syrup changed a lot compared to the control. Therefore it is amazing that the researchers conclude that the 75% replacement of sucrose with agave syrup has the same physical and  sensorial characteristics as the control.  

Here some notes:

The carbohydrate composition is given, but not the water content of the syrup? How the replacement was calculated as the recipe table shows a great amount of water added in the batters as well?

Additional information about the energy containing carbohydrate reduction compared with control and the agave syrup muffins would be appreciated as it was hypothesized in the aims to give a solution to the obesity problem.

row 168...: the viscosity decrease by addition of the syrup (contains water) and water might be more important than the small addition of pseudoplastic ? hydrocolloid for the viscosity

from row 181..: the oscillation results are explained wrong for the control batter; the figure 2;I showed that the G'<G'' at the low frequencies and G'>G''. You claimed that it was a solid at low frequencies and a liquid at high frequencies. 

row 201... The experiment of heating the batter during the rheological measurement was interesting.  The fact that the batter was thermosetting at higher temperature was important for the structure building of the muffins. This should be given much more thought and discussion! 

Figure 5: TPA results , it did not matter much which syrup or replacement level was used, all the syrup muffins differed from the control a lot.

Are the color results really so interesting that they need a huge table 2 and a lot of text? From row 372... is a good paragraph for the color.

Author Response

REVIEWER 2

This study was more like a product development work, how to replace sucrose levels with Agave syrup and with help of a hydrocolloid. The pre tests had been done as the levels of xanthan and agents were known. The characteristics of the muffins of Agave syrup changed a lot compared to the control. Therefore it is amazing that the researchers conclude that the 75% replacement of sucrose with agave syrup has the same physical and  sensorial characteristics as the control.  

Here some notes:

  1. COMMENT: The carbohydrate composition is given, but not the water content of the syrup? How the replacement was calculated as the recipe table shows a great amount of water added in the batters as well?

ANSWER: The moisture content of agave syrups provided by the supplier has been incorporated into the revised manuscript.

Revised manuscript (Lines 73‑78):

agave syrup (Mieles Campos Azules S. A. de C. V., Mexico; specifications of moisture and total sugars provided by the supplier: 23.20% moisture, 92.86% fructose, 0.15% glucose, 0.12% sucrose, 6.71% inulin, and 0.16% other carbohydrates); partially-hydrolyzed agave syrup (Mieles Campos Azules S. A. de C. V., Mexico; specifications of moisture and total sugars provided by the supplier: 23.30% moisture, 85.52% fructose, 0.40% glucose, 0.25% sucrose, 13.58% inulin, and 0.25% other carbohydrates)

The 30 g of water added in PHAS and AS batters was used to dissolve the 0.5 g xanthan gum, similarly to Martínez‑Cervera et al. (2012). In order to clarify the incorporation of this quantity of water into PHAS and AS batter formulations, a sentence has been added.

Revised manuscript (Lines 94‑96):

“First, the liquid ingredients (including 0.5 g of xanthan gum dissolved in 30 g of water), except for the sunflower oil, were introduced into the commercial kneader (Thermomix, TM31, Germany) and mixed for 1 min at a speed of 200 rpm.”

  1. COMMENT: Additional information about the energy containing carbohydrate reduction compared with control and the agave syrup muffins would be appreciated as it was hypothesized in the aims to give a solution to the obesity problem.

ANSWER: We agree with the reviewer. In order to avoid a misunderstanding in the aims of this work, the sentence has been removed and the corresponding information incorporated into the introduction.

Revised manuscript (Lines 56‑58; 483‑485):

“This natural sweetener, composed of fructose and fructooligosaccharides, has proven to have beneficial properties on human health, such as high prebiotic capacity, low glycemic index, and could prevent obesity and type II diabetes mellitus [16], [17]”.

  1. Hooshmand, S.; Holloway, B.; Nemoseck, T.; Cole, S.; Petrisko, Y.; Hong, M. Y.; Kern, M. Effects of Agave nectar versus sucrose on weight gain, adiposity, blood glucose, insulin, and lipid responses in mice. J. Med. Food 2014, 17 (9), 1017–1021. https://doi.org/10.1089/jmf.2013.0162.
  2. COMMENT: row 168...: the viscosity decrease by addition of the syrup (contains water) and water might be more important than the small addition of pseudoplastic? hydrocolloid for the viscosity

ANSWER: A sentence has been incorporated in the revised manuscript in order to clarify the effect of water content in batter viscosity.

Revised manuscript (Lines 171‑173):

“This behavior could be related to the moisture content of agave syrups (23.20% and 23.30% for AS and PHAS, respectively) which could have induced the decrease in AS and PHAS batter viscosities”.

  1. COMMENT: from row 181..: the oscillation results are explained wrong for the control batter; the figure 2;I showed that the G'<G'' at the low frequencies and G'>G''. You claimed that it was a solid at low frequencies and a liquid at high frequencies.

ANSWER: Thank you for pointing this out. There was actually a mistake in Figure 2.I., with the full and empty triangles erroneously switched. The figure has been corrected.

  1. COMMENT: row 201... The experiment of heating the batter during the rheological measurement was interesting.  The fact that the batter was thermosetting at higher temperature was important for the structure building of the muffins. This should be given much more thought and discussion! 

ANSWER: We agree with the reviewer. Your suggestions have been incorporated into the discussion section.

Revised manuscript (Lines 214‑216; 219-222; 506‑508):

The moisture content of the agave syrups could have contributed to higher values of the thermosetting temperature in replaced muffins than in the control since a higher water content needed to evaporate for the structure of the replaced muffins to be built.

In summary, the sucrose replacement by AS and PHAS and a combination of xanthan gum and doubled quantities of leavening agents increased the thermosetting temperature in batters. This fact is important for the correct formation of water vapor and CO2 in the batter, as well as their diffusion and expansion into occluded air cells during the baking process [3], [25].

  1. Alvarez, M. D.; Herranz, B.; Fuentes, R.; Cuesta, F. J.; Canet, W. Replacement of wheat flour by chickpea flour in muffin batter: effect on rheological properties. J. Food Process Eng. 2017, 40 (2), e12372. https://doi.org/10.1111/jfpe.12372.
  2. COMMENT: Figure 5: TPA results, it did not matter much which syrup or replacement level was used, all the syrup muffins differed from the control a lot.

ANSWER: A sentence has been added to the revised manuscript.

Revised manuscript (Lines 289‑292):

Since the textural properties of muffins depend greatly on their crumb structure, all samples with over 25% sucrose substitution by both AS and PHAS presented significant differences (p<0.05) with the control for all textural properties evaluated. As a result, all the replaced muffins considerably differed from the control regardless of syrup type or replacement level.

  1. COMMENT: Are the color results really so interesting that they need a huge table 2 and a lot of text? From row 372... is a good paragraph for the color.

ANSWER: We agree with the reviewer. The color section has been restructured and synthetized. Table 2 has been deleted.

Revised manuscript (Lines 342‑380):

“In general terms, the control muffin showed a golden‑brown crust and a yellow crumb, which are both characteristic of bakery products. However, as the percentage of sucrose substitution increased, muffin color in general tended to be darker than in control samples, with the muffin crust turning towards red hues and the crumb losing some of its yellowness.

The main effect of sucrose replacement on L* was significant (p<0.05) both in the muffin crust and crumb. As sucrose substitution increased, the L* values (ranging from 46.27 to 42.50 for the crust and from 61.34 to 53.23 for the crumb) significantly decreased (p<0.05) in comparison with the control samples (58.81 and 69.16 for the crust and crumb, respectively).

Regarding the C*ab of the crust, significant effects (p<0.05) were observed for sucrose replacement as well as for agave syrup type, with a significant interaction (p<0.05) between these factors. As for the crumb C*ab, only sucrose substitution had a significant effect (p<0.05). As sucrose replacement increased, the color intensity of both crust and crumb of the muffins showed different tendencies, whereas the crusts values decreased (40.59 for control and 35.99‑32.22 for replaced muffins), the crumb values increased (21.87 for control and 30.27‑27.50 for replaced muffins). Both presented significant differences (p<0.05) between control samples and the rest of the samples.

The main effect of sucrose substitution on h*ab was significant (p<0.05) on both the muffin crust and crumb. Replaced muffins exhibited lower h*ab values of both crumb and crust (ranging from 65.02 to 62.47 for the crust and from 84.07 to 76.10 for the crumb) than control samples (77.55 and 94.79 for the crust and crumb, respectively). Moreover, as percentages of sucrose replacement increased, the crumb h*ab values decreased. This shows that sucrose replacement by agave syrup provokes hue changes from yellow to orange‑red, both in muffin crumb and crust.

As for ΔE*, the main effect of two factors was significant (p<0.05) for the crust and the crumb. In general, the samples where sucrose was replaced with PHAS syrup showed higher ΔE* values in the crust (20.45), whereas crumb ΔE* values were similar for both AS and PHAS (ranging from 14.66 to 18.50). According to Bodart et al. [30], when the values obtained for total color difference are over three, the color variations between the analyzed samples are visible to the human eye. In this study, all samples with sucrose replacement showed ΔE* values of over three when compared to the control formulation, meaning the variations in color were easily noticeable to the naked eye.

The difference in color exhibited by the different formulations can mainly be attributed to the intensification of non‑enzymatic browning (Maillard reaction) that happened as the amount of incorporated agave syrup increased [31], [32]. Similar results were reported by Kocer et al. [33] where, as the amount of sucrose replaced by polydextrose increased, the baked products became darker. When substituting sucrose by other sweeteners such as erythritol, sorbitol, maltitol, xylitol, and mannitol, the opposite tends to occur as they do not contribute to the Maillard reaction [3], [34]. On a smaller scale, the muffin color was also affected by the dark color of the syrups in comparison to sucrose. Temperature fulfills an important role regarding the color of the crust and crumb of bakery products. Since Maillard reaction is temperature dependent, its effect will vary according to the maximum temperature reached. The crust tends to present a greater change due to the fact that it is exposed to higher temperatures, while the crumb exposition to the heat is more limited [32].”

Round 2

Reviewer 2 Report

The text has improved and the moisture issue (water addition) has been taken into account a bit more in the discussion.  Still the nutritional and energy calculations would be interesting, maybe in another study later!

There are still some discussion sentences that are weakly formulated like  "possibly, could" (171,173,194, 214 etc), using  "effect" and "affect", which showed lack of experience in scientific discussion.

Author Response

NOTE: All the corrections in the revised manuscript have been made in blue, in order to facilitate the final revision.

  1. COMMENT: The text has improved and the moisture issue (water addition) has been taken into account a bit more in the discussion. Still the nutritional and energy calculations would be interesting, maybe in another study later!

ANSWER: Thank you for your comment. It will be taken into account in future work.

  1. COMMENT: There are still some discussion sentences that are weakly formulated like "possibly, could" (171,173,194, 214 etc), using "effect" and "affect", which showed lack of experience in scientific discussion.

ANSWER: The discussion sentences that you mention are weakly formulated on purpose. The reason for this is that these are hypotheses or different possibilities for the explanation of our results. Where applicable, “could” was changed for “can”.

Revised manuscript (Lines 171-173; 194; 214-217; 416; 427):

- This behavior is quite possibly related to the moisture content of agave syrups (23.20% and 23.30% for AS and PHAS, respectively) which would have induced the decrease in AS and PHAS batter viscosities. Lines 171-173

- The competition for the available water between sucrose and xanthan gum has led to a ... Line 194

- Most probably, the moisture content of the agave syrups has contributed to higher values of the thermosetting temperature in replaced muffins than in the control since a higher water content needed to evaporate for the structure of the replaced muffins to be built. Lines 214-217

- Based on the results of our study, both AS and PHAS can be used as an alternative for sucrose... Line 416

- agave syrup can be a good alternative up to 75% of sucrose substitution. When 100% of sucrose was..Line 427
